# Context-dependent ciliary regulation of hedgehog pathway repression in tissue morphogenesis

Sun-Hee Hwang[1], Kevin Andrew White[1], Bandarigoda Nipunika Somatilaka[1,2], Baolin Wang[3], Saikat Mukhopadhyay[1]*

1 Department of Cell Biology, University of Texas Southwestern Medical Center, Dallas, Texas, United States of America, 2 Present address, Department of Dermatology, University of Texas Southwestern Medical Center, Dallas, Texas, United States of America, 3 Department of Genetic Medicine, Weill Medical College of Cornell University, New York, New York, United States of America

* saikat.mukhopadhyay@utsouthwestern.edu

**Data Availability Statement:** All relevant data are within the paper and its Supporting Information files.

## Abstract

A fundamental problem in tissue morphogenesis is identifying how subcellular signaling regulates mesoscale organization of tissues. The primary cilium is a paradigmatic organelle for compartmentalized subcellular signaling. How signaling emanating from cilia orchestrates tissue organization—especially, the role of cilia-generated effectors in mediating diverse morpho-phenotypic outcomes—is not well understood. In the hedgehog pathway, bifunctional GLI transcription factors generate both GLI-activators (GLI-A) and GLI-repressors (GLI-R). The formation of GLI-A/GLI-R requires cilia. However, how these counterregulatory effectors coordinate cilia-regulated morphogenetic pathways is unclear. Here we determined GLI-A/GLI-R requirements in phenotypes arising from lack of hedgehog pathway repression (derepression) during mouse neural tube and skeletal development. We studied hedgehog pathway repression by the GPCR GPR161, and the ankyrin repeat protein ANKMY2 that direct cAMP/protein kinase-A signaling by cilia in GLI-R generation. We performed genetic epistasis between *Gpr161* or *Ankmy2* mutants, and *Gli2/Gli3* knockouts, *Gli3R* knock-in and knockout of *Smoothened*, the hedgehog pathway transducer. We also tested the role of cilia-generated signaling using a *Gpr161* ciliary localization knock-in mutant that is cAMP signaling competent. We found that the cilia-dependent derepression phenotypes arose in three modes: lack of GLI-R only, excess GLI-A formation only, or dual regulation of either lack of GLI-R or excess GLI-A formation. These modes were mostly independent of Smoothened. The cAMP signaling-competent non-ciliary *Gpr161* knock-in recapitulated *Gpr161* loss-of-function tissue phenotypes solely from lack of GLI-R only. Our results show complex tissue-specific GLI-effector requirements in morphogenesis and point to tissue-specific GLI-R thresholds generated by cilia in hedgehog pathway repression. Broadly, our study sets up a conceptual framework for rationalization of different modes of signaling generated by the primary cilium in mediating morphogenesis in diverse tissues.

**Funding:** This study was supported by the National Institutes of Health grant R35GM144136 (SM) and R01GM140115 (BW). The content is solely the responsibility of the authors and does not necessarily represent the official views of the National Institutes of Health. The funders had no role in study design, data collection and analysis, decision to publish, or preparation of the manuscript.

**Competing interests:** The authors have declared that no competing interests exist.

## Author summary

The primary cilium is a minute cellular compartment that functions like an antenna in regulating cellular signaling. Defects in cilia and ciliary signaling disrupts tissue organization. However, how signaling generated by the cilia orchestrates tissue organization is not well understood. The hedgehog pathway is a developmental pathway that is dependent on cilia. In the hedgehog pathway, the GLI transcription factors generate mutually exclusive activators or repressors. The formation of both activators and repressors requires cilia. However, how cilia coordinate these factors in regulating tissue organization is unclear. Here, we studied defects in neural tube and skeletal development in the mouse that we described to result from lack of the ciliary repression of the hedgehog pathway. By studying whether the cilia generated activators or repressors regulated the tissue outcomes, we uncovered distinct modes by which ciliary outputs finally coordinated tissue organization. These modes showed to be from lack of the repressor only, excess activator only, or from dual regulation by lack of the repressor or activator. Lack of cilium specific trafficking also predominantly regulated tissue outcomes from lack of repressor only. Overall, our study uncovers general principles by which primary cilia organize tissue architecture.

## Introduction

Mesoscale organization of tissues require complex interplay between differentiation programs that give rise to different cell types [1]. The primary cilium is a paradigmatic organelle for compartmentalized subcellular signaling during morphogenesis [2]. Cilia are widely present and can transduce cellular response to extracellular signals such as hedgehog (HH) morphogens [3]. Disruption of cilia causes phenotypes in multiple tissues [4]. There has been a plethora of studies showing signaling at the level of cilia [5]. However, how signaling from cilia orchestrates morphogenesis in diverse tissues and in different cell types in the same tissue is not well understood (**Fig 1A**). Especially, the role of ciliary effectors that function as intermediaries between cilia and downstream pathways in tissue morphogenesis is not clear. In addition, signaling outputs from cilia often have counter-regulatory functions, such as in the HH pathway [6] or during kidney tubular homeostasis [7]. It is unclear if such counter-regulatory functions of cilia balance each other in regulating downstream outputs or if the cilium sets thresholds for the effectors in determining morpho-phenotypic outcomes. Finally, it is important to infer general principles by which cilia direct morphogenesis in diverse tissues.

We study HH signaling regulation by cilia in diverse tissues to answer these fundamental questions. HH signaling outputs are regulated by bi-functional GLI transcriptional factors that can function as both activators and repressors (**Fig 1A**). Post-translational changes in GLI2 and GLI3 induce formation of the repressor and activator [8–16]. The cilium is required in forming both GLI repressor and activator during development [3]. GLI3 is the major repressor, and GLI2 is the major activator, but both GLI2/3 can form either repressor or activator. Furthermore, as these counterregulatory factors are generated from the same parental bifunctional GLI2/3 transcription factor, lack of a repressor (derepression) might tip the balance towards generation of the activator [6,17]. Disruption of cilia or *Gli2/3* knockout removes both activator and repressor and is not suitable for uncovering the role of these counter-regulatory effectors. For e. g., phenotypes arising from a loss of GLI-R, could result in a corresponding rise of GLI-A [6]. If rise in GLI-A rather than loss of GLI-R results in the phenotype, it would not be apparent from *Gli2/3* knockouts or lack of cilia, as they cause a loss of both GLI-A and GLI-R. Therefore, to uncover the precise roles of the GLI effectors, models affecting HH

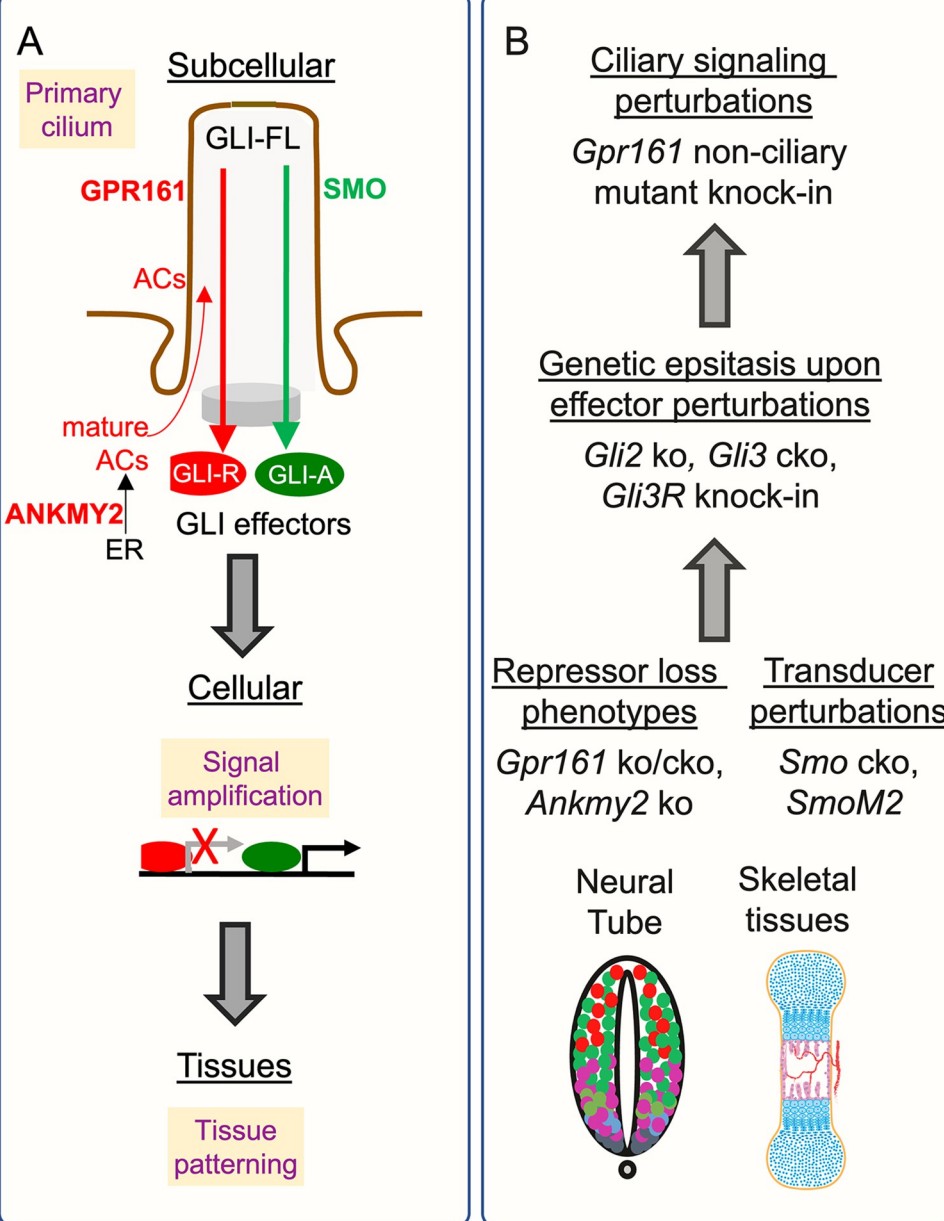

**Fig 1. Testing ciliary effectors of HH signaling in uncovering distinct tissue-specific modes of regulation. (A)** Gaps in our understanding on how signaling from cilia regulates morphogenesis in diverse tissues and in different cell types in the same tissue. We utilized two models of cilia regulated HH-pathway repression while maintaining intact cilia. The models include two key repressors of the hedgehog pathway that we described, the cilia localized GPCR GPR161, and the ankyrin repeat protein ANKMY2. **(B)** Bottom-up approach to understand role of cilia generated effectors in two distinct paradigms of morphogenesis: neural tube and skeletal development. We performed genetic epistasis between *Gpr161/Ankmy2* mutants and *Gli2/Gli3* ko, *Gli3R* knock-in and *Smo* ko. Role of GPR161 ciliary signaling was compared using a *Gpr161* cilia localization mutant with respect to *Gpr161* cko. Abbreviations: ko, knockout; cko; conditional knockout; ER, endoplasmic reticulum.

pathway repression, selectively perturbing GLI-R roles, and testing cilia generated signaling without disrupting cilia are required.

Here we utilized two models of cilia regulated HH-pathway repression while maintaining intact cilia in distinct morphogenetic programs: neural tube and skeletal development

(**Fig 1A**). The models include two key repressors of the HH pathway we described before, the cilia localized GPCR GPR161 [18], and the ankyrin repeat protein ANKMY2 [17]. Both determine cAMP-protein kinase A signaling by cilia in GLI-R generation (**Fig 1A**). The lack of GLI-R formation in *Ankmy2* and *Gpr161* mutants provided a unique opportunity to test the critical role of cilia mediated GLI effectors in morphogenesis while maintaining intact cilia. The tissues were chosen based on phenotypes resulting from HH pathway derepression that we originally described and found to be fully cilia dependent [17–19]. The severity of some of these phenotypes paralleled lack of the HH receptor Patched emphasizing role of HH repression in tissue morphogenesis.

To test the dependance of neural tube and skeletal phenotypes on cilia dependent GLI effectors, we performed genetic epistasis between *Gpr161/Ankmy2* mutants and *Gli2/Gli3* ko, *Gli3R* knock-in [20] and *Smoothened (Smo)* ko. We also tested the role of cilia-generated GPR161 signaling using a *Gpr161* ciliary localization mutant [21] (**Fig 1B**). Our results show that HH pathway derepression causes phenotypes as strong as HH pathway activation and are primarily independent of SMO [17]. Testing the dependance of these phenotypes on GLI effectors show that they function in three modes: solely activation, lack of repression or dual regulation. Lack of cilium specific GPR161 signaling manifest phenotypes that are predominantly dependent on GLI repression. In sum, our results set up a conceptual framework for rationalization of different modes of cilia-mediated GLI-A/R regulation in diverse tissues and point to a role of tissue specific GLI-R thresholds established by cilia in HH pathway repression.

## Results

### Testing differential requirements for GLI3R and GLI2 in neural tube patterning to study ciliary effectors in morphogenesis

To test the role of cilia regulated HH-pathway repression in morphogenesis, we first utilized the well-studied model of neural tube development. The neuroprogenitors acquire different spatial identities along the dorso-ventral axis due to a complex interplay of transcription factors, which are expressed in response to relative variations in sonic hedgehog (SHH) levels secreted from notochord, as well as the duration for which they are exposed [22]. The neuroprogenitors are ciliated; HH-dependent ventral patterning is disrupted in progenitors without cilia [23].

Ventral-most neural tube patterning is dependent on the downstream activation of GLI2 [24–26] and activator function of GLI3 [26], whereas intermediate-level patterning is GLI3R regulated [25,27–29]. As both ANKMY2 and GPR161 regulate GLI-R formation, lack of GLI-R in these mutants can result in increased HH signaling. Importantly, lack of either causes ventral cell types to be ectopically specified at the expense of lateral and dorsal cell types (ventralization) [17,18], and such ventralization occurs independent of SMO [17]. Thus GLI-R mediated repression seems to promote SHH-dependent ventral patterning.

To test regulation of GLI-R in specification and expansion of floor plate and other ventral most progenitors, we compared the differential requirements for GLI3R and GLI2 in *Ankmy2* and *Gpr161* knockout (ko) embryos. We tested the role of GLI2 using a *Gli2* ko and GLI3R by using a knock-in allele ($GLI3^{\Delta701}$) [20]. Unlike the commonly used GLI3R allele ($GLI3^{\Delta699}$) [30], *Prx1-Cre* expressing embryos homozygous for the conditional $GLI3^{\Delta701}$ allele resembles *Shh* ko during limb development [20], as expected from GLI3R antagonizing SHH [31,32]. Thus, the $GLI3^{\Delta701}$ allele generates a more potent repressor than the $GLI3^{\Delta699}$ allele [20]. The comparative approach directly targeted the role of cilia generated effectors in SHH pathway repression during dorsoventral neural tube patterning.

## *Gli3R* expression restored both high and low HH signaling dependent progenitors in *Ankmy2* knockout

We first investigated if the *Ankmy2* ko neural tube phenotype is dependent on GLI2 and/or lack of GLI3R by generating *Ankmy2; Gli2* double ko and *Ankmy2* ko; *GLI3R^{Δ701/+}* embryos. We found that the *Ankmy2* ko; *Gli3R^{Δ701/+}* embryos persisted till at least 22–24 somite stage and were well turned (**S1 Fig**). In contrast, *Ankmy2* ko littermate embryos arrested by 16 somites and had turning defects. Similarly, the *Ankmy2; Gli2* double ko survived till the 26–28 somite stage [17].

When dissected at E9.25, we noted extensive ventralization in the *Ankmy2* ko neural tube as reported earlier [17]. Specifically, floor plate progenitors expressing FOXA2, p3 progenitors expressing NKX2.2, pMN progenitors expressing OLIG2, and p3/pMN/p2 progenitors expressing NKX6.1 showed enlarged expression domains that expanded often fully into dorsal regions throughout most of the spinal cord (**Fig 2A–2A''' and 2B–2B'''**). In contrast, the dorsolateral neural tube marker PAX6 was completely absent in the *Ankmy2* ko (**Fig 2A and 2B**). Such ventralization was similar in extent and severity to HH receptor *Ptch1* knockout and was fully cilia dependent at all rostrocaudal levels [17].

In contrast, at the thoracic level of the *Ankmy2* ko; *GLI3R^{Δ701/+}* embryos by 22–24 somites, the neural tube was closed with FOXA2 and NKX2.2 progenitors as seen in wild type with only very limited ventralization (**Fig 2A, 2A' and 2A''**), but not completely reduced as in *Ankmy2; Gli2* double ko or *Gli2* single ko (**Fig 2A, 2A' and 2A''**). There was some persistent but limited ventralization of the NKX6.1 domain in the *Ankmy2* ko; *GLI3R^{Δ701/+}* as seen in the *Ankmy2; Gli2* double ko, whereas OLIG2 domain expansion was less affected (**Fig 2A and 2A'''**). PAX6 was restored in the *Ankmy2* ko; *GLI3R^{Δ701/+}* compared to *Ankmy2* single ko but still restricted dorsolaterally compared to wild type embryos (**Fig 2A**).

At the lumbar level of the *Ankmy2* ko; *GLI3R^{Δ701/+}* embryos, FOXA2 and NKX2.2 progenitor domains were as in wild type, but not reduced as in *Ankmy2; Gli2* double ko or *Gli2* single ko (**Fig 2B, 2B' and 2B''**). OLIG2 progenitors were also restored like wild type domains. There was some persistent but limited ventralization of NKX6.1 domain as seen in *Ankmy2; Gli2* double ko. PAX6 was restored like wild type. *GLI3R* by itself also caused partial reduction of FOXA2 and NKX2.2 progenitors compared to wild type but less than *Gli2* ko (**Fig 2B–2B'''**).

The delayed embryonic lethality in both *Ankmy2* ko; *Gli3R^{Δ701}/+* and *Ankmy2; Gli2* double ko occurred despite both exhibiting open hind brain (exencephaly) similar to *Ankmy2* ko (**S1 Fig**). At the hindbrain levels of the *Ankmy2* ko; *GLI3R^{Δ701}/+* embryos, the hindbrain showed ventralized NKX6.1 like *Ankmy2* ko. However, unlike *Ankmy2* ko, FOXA2 levels were reduced, while PAX6 was partially restored, as seen in *Ankmy2; Gli2* double ko (**S1 Fig**).

To summarize, both FOXA2 and NKX2.2 expressing neuroprogenitors were repressed beyond the floor plate and p3 domains of the neural tube by either *Gli3R* expression or from GLI2 loss, with the lumbar levels showing more GLI3R dependence than the rostral levels (**Fig 2C**). NKX6.1 dorsal expansion was also partially overcome in the presence of GLI3R or in the absence of GLI2. Overall, most of the ventralization and neural tube closure defects in the *Ankmy2* ko, except exencephaly, were overcome in the presence of GLI3R or in the absence of GLI2.

## *Gli3R* expression restored both high and low HH signaling dependent progenitors in *Gpr161* knockout

We next investigated another model of HH pathway derepression, the *Gpr161* ko neural tube, for dependance on GLI2 and/or lack of GLI3R. We generated *Gpr161; Gli2* double ko and *Gpr161* ko; *GLI3R^{Δ701/+}* embryos. As *Gpr161/Gli2* genes are on the same chromosome, we

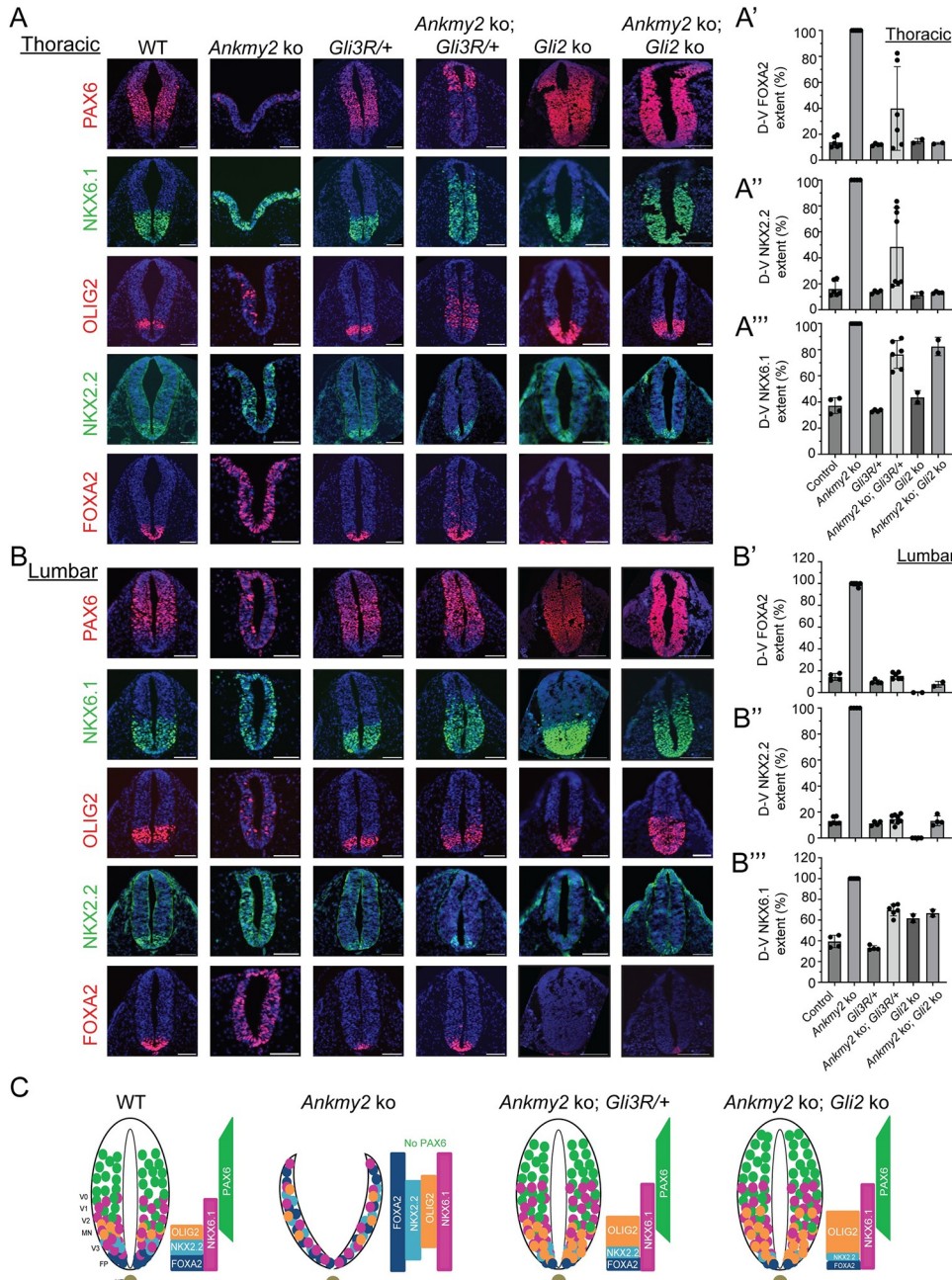

**Fig 2. *Gli3R* expression restored both high and low HH signaling dependent progenitors in *Ankmy2* ko. (A-B)** Panels show thoracic (A) and lumbar (B) neural tube horizontal sections immunostained using designated markers for embryos dissected at E9.25 of the following genotypes: wild-type (n = 5), *Ankmy2* ko (n = 5), *Gli3R/+* (n = 4), *Ankmy2* ko; *Gli3R/+* (n = 4), *Gli2* ko (n = 3), *Ankmy2* ko; *Gli2* ko (n = 3). All images are counterstained with DAPI. Quantification of thoracic (A'-A''') and lumbar (B'-B''') regions are shown. **(C)** Cartoon showing that strong ventralization in the spinal cord region of *Ankmy2* ko is dependent on both loss of GLI3R and GLI2 presence. The neuro progenitor domains (floor plate, p3, pMN, p2, p1, and p0) generate distinct neuronal subtypes (V3, MN, V2, V1, V0). FP progenitors express FOXA2, p3 progenitors express NKX2.2, pMN progenitors express OLIG2, and p3/pMN/p2 progenitors express NKX6.1 in wild-type. Abbreviations: FP, floor plate; NT, notochord; WT, wild-type. Scale: 100 μm. See also S1 Fig.

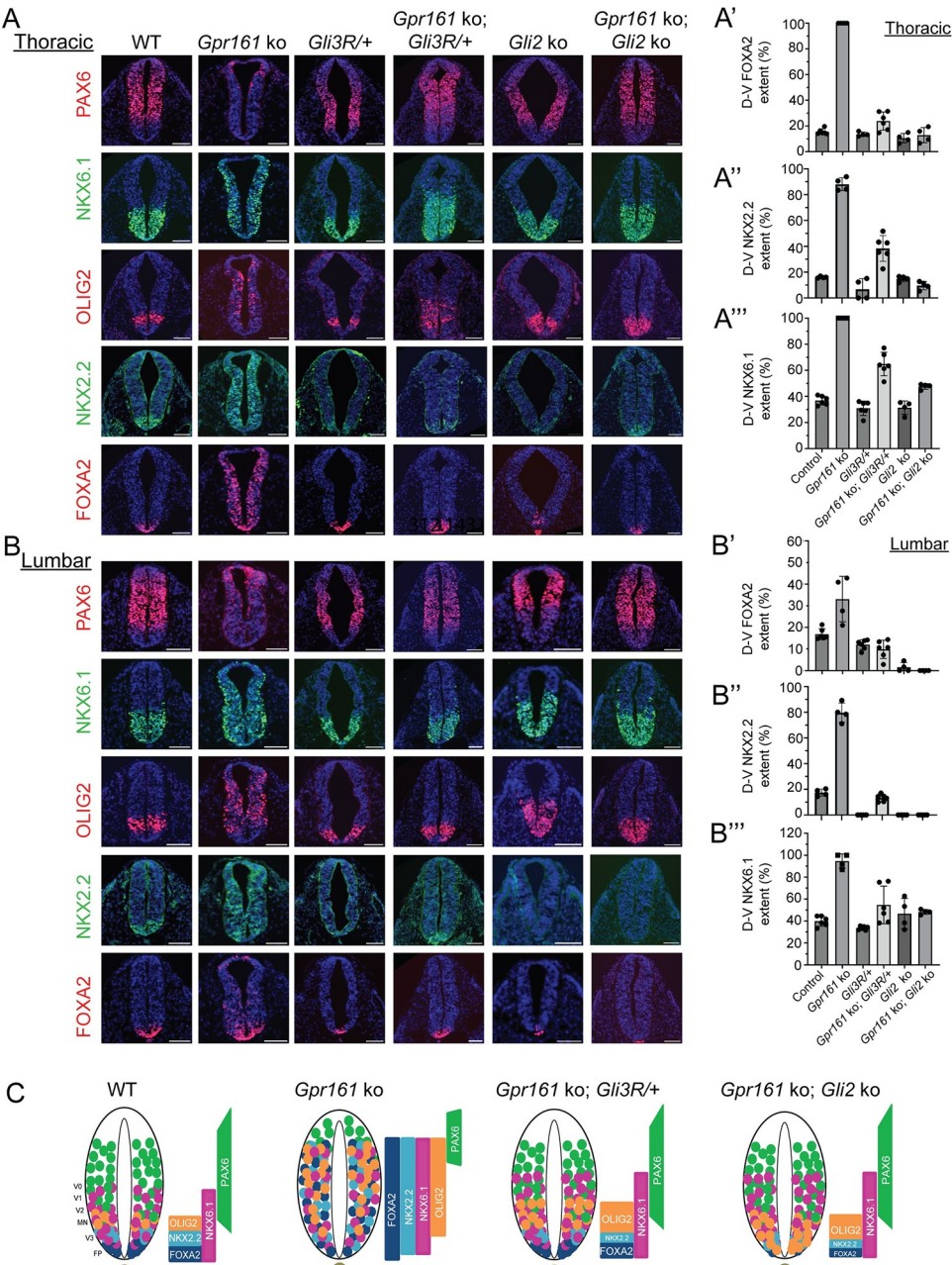

**Fig 3.** *Gli3R* **expression restored both high and low HH signaling dependent progenitors in** *Gpr161* **ko. (A-B)** Panels show thoracic (A) and lumbar (B) neural tube horizontal sections immunostained using designated markers for embryos dissected at E9.5 of the following genotypes: wild-type (n = 6), *Gpr161* ko (n = 3), *Gli3R/+* (n = 6), *Gpr161* ko; *Gli3R/+* (n = 6), *Gli2* ko (n = 6), *Gpr161* ko; *Gli2* ko (n = 6). All images are counterstained with DAPI. Quantification of thoracic (A"-A"') and lumbar (B'-B"') regions are shown. **(C)** Cartoon showing that the partial ventralization in the spinal cord region of *Gpr161* ko is dependent on both GLI3R loss and GLI2 presence. The neuronal subtype domains are depicted as in Fig 2C. Abbreviations: FP, floor plate; NT, notochord; WT, wild-type. Scale: 100 μm.

generated linked double mutants by recombination. Both *Gpr161* ko; *Gli3R*$^{\Delta701/+}$ and *Gpr161; Gli2* double ko survived till E12.75, past the embryonic lethality period for *Gpr161* ko at E10.5.

As we had demonstrated before [18], partial ventralization occurred in the *Gpr161* ko neural tube at E9.5 (**Fig 3A and 3B**), but to lesser extent as seen in *Ankmy2* ko (**Fig 2**). Specifically,

FOXA2, NKX2.2, OLIG2, and NKX6.1 showed enlarged expression domains, expanding into comparatively more dorsal regions throughout the rostrocaudal extent of the spinal cord (**Fig 3A–3A''' and 3B–3B'''**). PAX6 was expressed in dorsally restricted domains (**Fig 3A and 3B**). Such partial ventralization is fully cilia-dependent at all levels [18].

In contrast, at the thoracic level of the *Gpr161* ko; *Gli3R*$^{\Delta701/+}$ embryos by 22 somites, the neural tube had FOXA2 and NKX2.2 progenitors as seen in wild type (**Fig 3A, 3A' and 3A''**), but not completely reduced as in *Gpr161; Gli2* double ko or *Gli2* single ko (**Fig 3A, 3A' and 3A''**). OLIG2 progenitors were also partially restored, whereas *Gpr161; Gli2* double ko showed OLIG2 expansion to the floor plate (**Fig 3A**). There was partial restoration of NKX6.1 domain in the *Gpr161* ko; *Gli3R*$^{\Delta701/+}$ embryos like *Gpr161; Gli2* double ko (**Fig 3A and 3A'''**). Dorso-lateral expression of PAX6 was also partially restored in the *Gpr161* ko; *Gli3R*$^{\Delta701/+}$ embryos compared to *Gpr161* single ko like *Gpr161; Gli2* double ko (**Fig 3A**).

At the lumbar level of the *Gpr161* ko; *Gli3R*$^{\Delta701}$/+ embryos, the neural tube showed FOXA2 and NKX2.2 progenitors as seen in wild type, but not completely reduced as in *Gpr161; Gli2* double ko or *Gli2* single ko (**Fig 3B, 3B' and 3B''**). OLIG2 progenitors were also restored in the *Gpr161* ko; *Gli3R*$^{\Delta701/+}$ like wild type domains, whereas *Gpr161; Gli2* double ko showed OLIG2 expansion to the floor plate (**Fig 3B**). The NKX6.1 domain in *Gpr161* ko; *Gli3R*$^{\Delta701/+}$ was almost restored with some limited ventralization like *Gpr161; Gli2* double ko (**Fig 3B and 3B'''**). The dorsolateral marker PAX6 was restored in *Gpr161* ko; *Gli3R*$^{\Delta701}$/+ and *Gpr161; Gli2* double ko like wild type, compared to *Gpr161* single ko (**Fig 3B**).

Overall, these results suggest that partial ventralization in most of the dorsal spinal cord region of *Gpr161* ko for both high HH signaling progenitors (FOXA2, NKX2.2, OLIG2) and low HH signaling progenitors (NKX6.1) is dependent on both GLI3R loss and GLI2 presence (**Fig 3C**), with the lumbar levels showing more dependence for GLI3R than rostral levels.

## Testing differential requirements for GLI3R and GLI2 in skeletal development to study ciliary effectors in morphogenesis

To test the role of cilia regulated HH-pathway repression in morphogenesis, we also utilized skeletal development in fore limb long bones and calvarium as paradigms for studying derepression phenotypes. Skeletal morphogenesis occurs by endochondral or intramembranous ossification, based on dependance or lack of dependance on an intermediate cartilaginous template, respectively [33,34]. During endochondral ossification, periarticular chondrocytes differentiate into columnar, prehypertrophic and hypertrophic chondrocytes. IHH is secreted from prehypertrophic chondrocytes, and activates chondrocyte proliferation, differentiation of proliferating to hypertrophic chondrocytes and osteoblast differentiation in the perichondrium [33]. The chondrocytes are ciliated at all these stages [19,35].

The role of GLI2 and GLI3 has been studied in the context of IHH signaling. GLI3R is thought to prevent *PthrP* expression in the periarticular chondrocytes by IHH [36,37], whereas GLI2A is thought to regulate IHH-dependent preosteoblast to osteoblast differentiation [38]. However, we previously showed a complete lack of mineralization in the forearm long bones of *Prx1-Cre; Gpr161*$^{f/f}$ embryos [19]. The chondrocytes did not progress beyond the periarticular chondrocyte stage, were ciliated, and showed no IHH expression. We also demonstrated lack of calvarium intramembranous bone formation [19]. GPR161 localizes to cilia of mesenchymal stem cell lines that have potential for chondrogenic and osteogenic differentiation [39], but the exact cell type(s) affected from *Gpr161* loss *in vivo* in manifestation of these tissue phenotypes are not currently known. However, both these phenotypes were rescued from concomitant lack of cilia.

The limb and skeletal phenotypes in *Prx1-Cre; Gpr161*$^{f/f}$ embryos are not well understood and provided understudied yet important models for studying ciliary effectors during

morphogenesis. Here we compared the requirements for GLI3R, GLI3, GLI2, SMO, and ciliary localization of GPR161 in skeletal phenotypes of *Prx1-Cre; Gpr161$^{f/f}$* using *Gli3/Gli3R* conditional, *Gli2 ko* and *Gpr161* knock-in alleles. *Gli3$^{Δ701C}$* refers to the conditional *Gli3$^{Δ701}$* allele and is annotated as *Gli3R$^{f}$* in the subsequent figures for simplicity.

## GLI2/GLI3R dependance on lack of forelimb long bone mineralization in *Prx1-Cre; Gpr161$^{f/f}$*

We compared the mineralization of forearm long bones at E18.5 by skeletal staining with Alizarin Red (stains mineralized cartilage/bone) and Alcian Blue (stains unmineralized cartilage) (**Figs 4 and S2**). We also compared the lengths of the long bone primordia and the extent of mineralization with respect to the lengths. *Prx1-Cre; Gpr161$^{f/f}$* embryos showed a lack of mineralization in the forearm long bones, as we had reported before [19] (**Figs 4A, 4B, 4K and S2**). The length of humerus was also significantly reduced in *Prx1-Cre; Gpr161$^{f/f}$* embryos compared to wild type (**Figs 4A, 4B, 4K and S2**). These phenotypes were rescued from concomitant lack of cilia in *Prx1-Cre; Gpr161$^{f/f}$; Ift88$^{f/f}$* embryos (**Figs 4C–4D, 4K and S2**). In contrast, *Prx1-Cre; Gpr161$^{f/f}$; Gli2 ko* embryos showed rescue of lack of mineralization in radius and ulna but not that of humerus, compared to *Prx1-Cre; Gpr161$^{f/f}$* (**Figs 4E–4F and S2**). The radius and ulna in *Prx1-Cre; Gpr161$^{f/f}$; Gli2 ko* were of length like that of wild type or *Gli2 ko* alone, whereas the length of humerus in *Prx1-Cre; Gpr161$^{f/f}$; Gli2 ko* were only partially restored compared to wild type or *Gli2 ko* alone (**Figs 4B, 4E, 4F and S2**). The *Prx1-Cre; Gpr161$^{f/f}$; Gli3$^{Δ701C/+}$* embryos had restored mineralization in humerus, radius and ulna compared to *Prx1-Cre; Gpr161$^{f/f}$* (**Figs 4B, 4E, 4F and S2**). The forearm long bones in *Prx1-Cre; Gpr161$^{f/f}$; Gli3$^{Δ701C/+}$* also looked equivalent to *Prx1-Cre; Gli3$^{Δ701C/+}$* embryos lengthwise (**Figs 4G, 4H and S2**). The forearm long bones in *Prx1-Cre; Gpr161$^{f/f}$; Gli3$^{f/f}$* embryos showed lengths equivalent to that of *Prx1-Cre; Gpr161$^{f/f}$* unlike *Prx1-Cre; Gli3$^{f/f}$* that were unaffected (**Figs 4I, 4J and S2**). Mineralization in radius/ulna was lacking in *Prx1-Cre; Gpr161$^{f/f}$; Gli3$^{f/f}$* compared to *Prx1-Cre; Gpr161$^{f/f}$; Gli3$^{Δ701C/+}$* or *Prx1-Cre; Gpr161$^{f/f}$; Gli2 ko* that were less to not affected (**Figs 4G–4K and S2**). There was some limited mineralization in the small humerus in *Prx1-Cre; Gpr161$^{f/f}$; Gli3$^{f/f}$*, but it looked morphologically similar to that of *Prx1-Cre; Gpr161$^{f/f}$* embryos (**Figs 4I–4K and S2**).

  *Prx1-Cre; Gli3$^{f/f}$* and *Prx1-Cre; Ift88$^{f/f}$* resulted in preaxial polydactyly (duplicated first digit) likely from lack of GLI3R [35,40], whereas the *Prx1-Cre; Gli3$^{Δ701C/+}$* autopods showed only the first few digits resembling *Shh* ko [20]. *Prx1-Cre; Gli2 ko* autopods resembled wild type. In contrast, concomitant deletion of *Gli2, Gli3* or *Ift88* or expression of *Gli3R* in *Prx1-Cre; Gpr161$^{f/f}$* maintained the polysyndactyly in the latter [19], suggesting complex coregulation of GLI effectors as reported earlier in hindlimb autopods (**S3 Fig**) [40].

  The scapular blade and scapular spine were affected in *Prx1-Cre; Gpr161$^{f/f}$* embryos (**Fig 4B**). The scapular blade and scapular spine dysmorphogenesis in *Prx1-Cre; Gpr161$^{f/f}$; Gli3$^{f/f}$* (**Fig 4J**) but was partially restored in *Prx1-Cre; Gpr161$^{f/f}$; Gli2 ko*, and less so in *Prx1-Cre; Gpr161$^{f/f}$; Gli3$^{Δ701C/+}$* embryos (**Fig 4F and 4H**).

  To summarize, concomitant GLI2 lack or *Gli3R* expression both restored mineralization in *Prx1-Cre; Gpr161$^{f/f}$* radius and ulna. GLI2 lack in *Prx1-Cre; Gpr161$^{f/f}$* was more effective in restoring radius and ulna length and morphology than *Gli3R* expression. In contrast, concomitant expression of *Gli3R* specifically restored mineralization in *Prx1-Cre; Gpr161$^{f/f}$* humerus.

## GLI2/GLI3R dependance on lack of osteogenesis in *Prx1-Cre; Gpr161$^{f/f}$* calvaria

We next compared the osteogenesis of calvarium at E18.5 for the different mutants (**Fig 5**). *Prx1-Cre; Gpr161$^{f/f}$* embryos lacked posterior calvarium mineralization and osteogenesis, as we

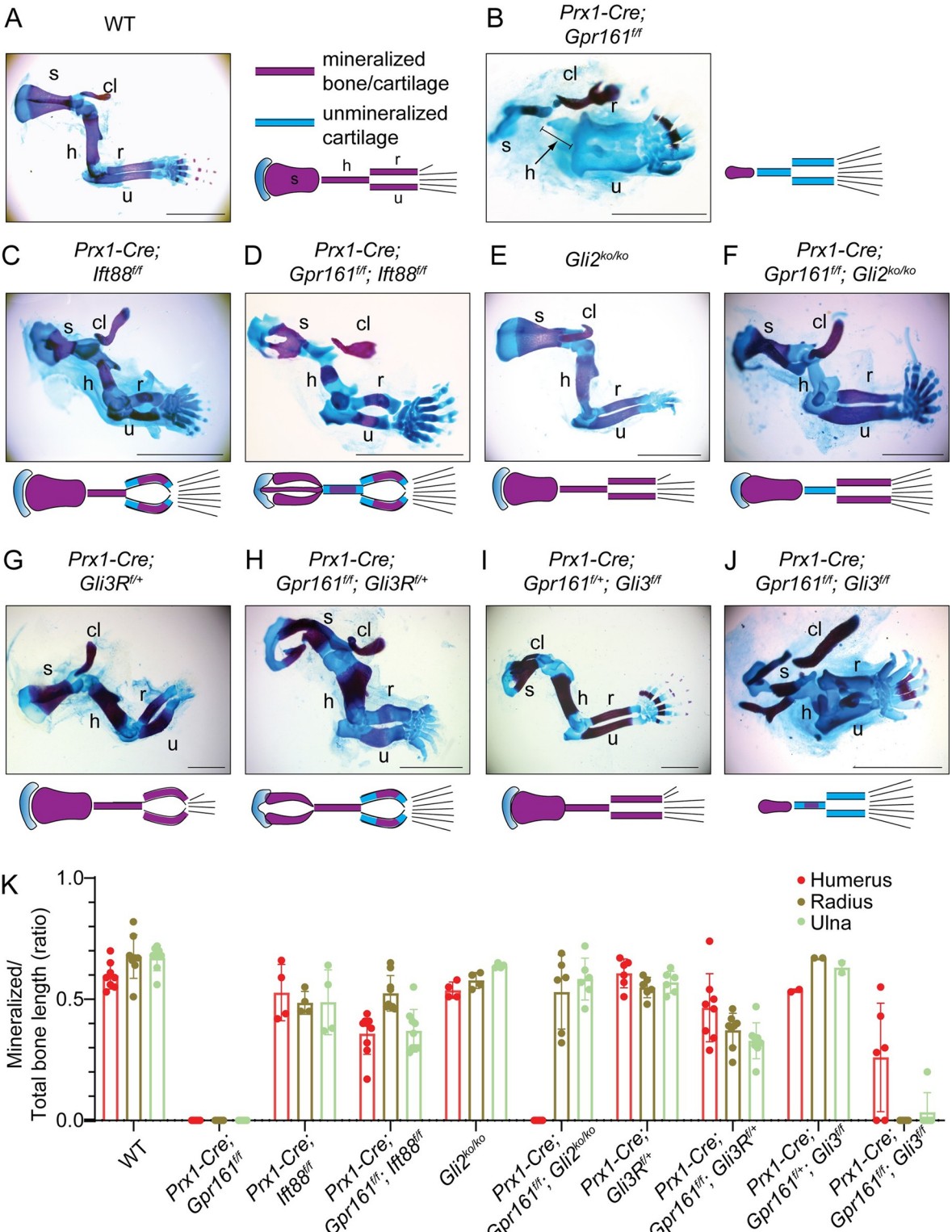

**Fig 4. GLI2/GLI3R dependance on forelimb long bone mineralization in *Prx1-Cre; Gpr161^{f/f}*. (A-J)** Alcian blue (unmineralized cartilage) and alizarin red (mineralized cartilage and bone) staining of forelimb in E18.5 embryos. The genotypes shown are as follows: (A) *Prx1-Cre; Gpr161^{f/+}* (WT, n = 15), (B) *Prx1-Cre; Gpr161^{f/f}* (n = 9), (C) *Prx1-Cre; Ift88^{f/f}* (n = 5), (D) *Prx1-Cre; Ift88^{f/f}; Gpr161^{f/f}* (n = 6), (E) *Gli2* ko (n = 4), (F) *Prx1-Cre; Gpr161^{f/f}, Gli2* ko (n = 3), (G) *Prx1-Cre; Gli3R^{f/+}* (n = 7), (H) *Prx1-Cre; Gpr161^{f/f}; Gli3R^{f/+}* (n = 4), (I) *Prx1-Cre; Gpr161^{f/+}, Gli3^{f/f}* (n = 5), (J) *Prx1-Cre; Gpr161^{f/f}; Gli3^{f/f}* (n = 3). **(K)** Quantification of mineralized to total bone primordia length

ratios shown. Each data point represents individual bone primordia quantified. Abbreviations: clavicle, cl; scapula, s; h, humerus; r, radius; u, ulna. Scale, 2 mm (A-J). See also S2 Fig for quantification of forearm lengths and S3 Fig showing autopods.

had reported before [19]. The lack of mineralization and intramembranous bone formation was seen over most of the posterior frontal, parietal and interparietal bones (**Fig 5A and 5B**). These phenotypes were mostly rescued from concomitant lack of cilia in *Prx1-Cre; Gpr161^{f/f}; Ift88^{f/f}* embryos, except some lack of mineralization in the lateral section of the parietal bone (**Fig 5C, 5D and 5D'**). We found that *Prx1-Cre; Gpr161^{f/f}; Gli2* ko embryos restored the lack of mineralization in frontal and interparietal bones but not that of the parietal bone, compared to *Prx1-Cre; Gpr161^{f/f}* (**Fig 5E, 5F and 5F'**). The *Prx1-Cre; Gpr161^{f/f}; Gli3^{Δ701C/+}* embryos had restored mineralization in frontal, medial parietal, and interparietal regions compared to *Prx1-Cre; Gpr161^{f/f}* (**Fig 5G, 5H and 5H'**). The *Prx1-Cre; Gpr161^{f/f}; Gli3^{f/f}* embryos looked comparable to *Prx1-Cre; Gpr161^{f/f}* in their lack of mineralization in posterior calvarium (**Fig 5I and 5J**).

Overall, concomitant GLI2 lack or *Gli3R* expression both restored mineralization and osteogenesis in posterior frontal and interparietal regions of *Prx1-Cre; Gpr161^{f/f}*. In contrast, concomitant *Gli3R* expression specifically restored mineralization and osteogenesis in the medial parietal regions of *Prx1-Cre; Gpr161^{f/f}* calvarium.

## Forelimb long bone and calvarial osteogenesis in *Prx1-Cre; Gpr161^{f/f}* was mostly Smoothened independent

SMO, the transducer of the HH pathway, is required for GLI2 activation [6]. We previously showed that neural tube ventralization in *Gpr161* ko is mostly independent of SMO. However, expansion of floorplate progenitors that require high level of SHH are partially SMO dependent [18]. In contrast, the ventralization in *Ankmy2* ko neural tube, even that of the floor plate neuroprogenitors, is fully SMO-independent [17]. We thus tested the SMO dependance on *Prx1-Cre; Gpr161^{f/f}* skeletal phenotypes. Loss of SMO in *Prx1-Cre; Smo^{f/f}* caused extremely short but mineralized long bones in forearm (**Figs 6A, 6E, 6M and S4**). In contrast, constitutively active *SmoM2* expression caused complete lack of mineralization of humerus, and only partial to no mineralization of the radius and ulna (**Figs 6B, 6F, 6M and S4**). However, *Prx1-Cre; Gpr161^{f/f}; Smo^{f/f}* embryos showed overall lack of mineralization in forearm long bones like *Prx1-Cre; Gpr161^{f/f}* except central most mineralization in radius and ulna (**Figs 6C, 6D, 6G, 6H, 6M and S4**). The forelimb autopod of *Prx1-Cre; Smo^{f/f}* embryos showed only the first few digits as expected from lack of SHH mediated signaling [20], whereas constitutively active *SmoM2* expression showed polysyndactyly similar to *Prx1-Cre; Gpr161^{f/f}* (**Figs 6E–6G and S3**). However, the forelimb autopod of the *Prx1-Cre; Gpr161^{f/f}; Smo^{f/f}* embryos was like that of *Prx1-Cre; Gpr161^{f/f}* but unlike *Prx1-Cre; Smo^{f/f}* (**Figs 6H and S3**). Similarly, lack of calvarium mineralization was observed in *Prx1-Cre; Gpr161^{f/f}; Smo^{f/f}* like *Prx1-Cre; Gpr161^{f/f}* and unlike *Prx1-Cre; Smo^{f/f}* that showed calvaria mineralization (**Fig 6I–6L**).

Overall, lack of mineralization and osteogenesis phenotypes in forearm long bones and calvarium of *Prx1-Cre; Gpr161^{f/f}* were mostly SMO independent.

## Lack of GPR161 from cilia manifest phenotypes predominantly mediated from GLI3R loss

We recently generated a non-ciliary yet cAMP signaling-competent *Gpr161* allele (*Gpr161^{mut1}*) [21]. As GPR161 lack causes both GLI-R loss and GLI-A generation [21], we tested how the specific lack of GPR161 from cilia correlated with the conditional ko phenotypes in skeletal morphogenesis (**Fig 7**). We found that *Prx1-Cre; Gpr161^{mut1/f}* embryos showed mineralization

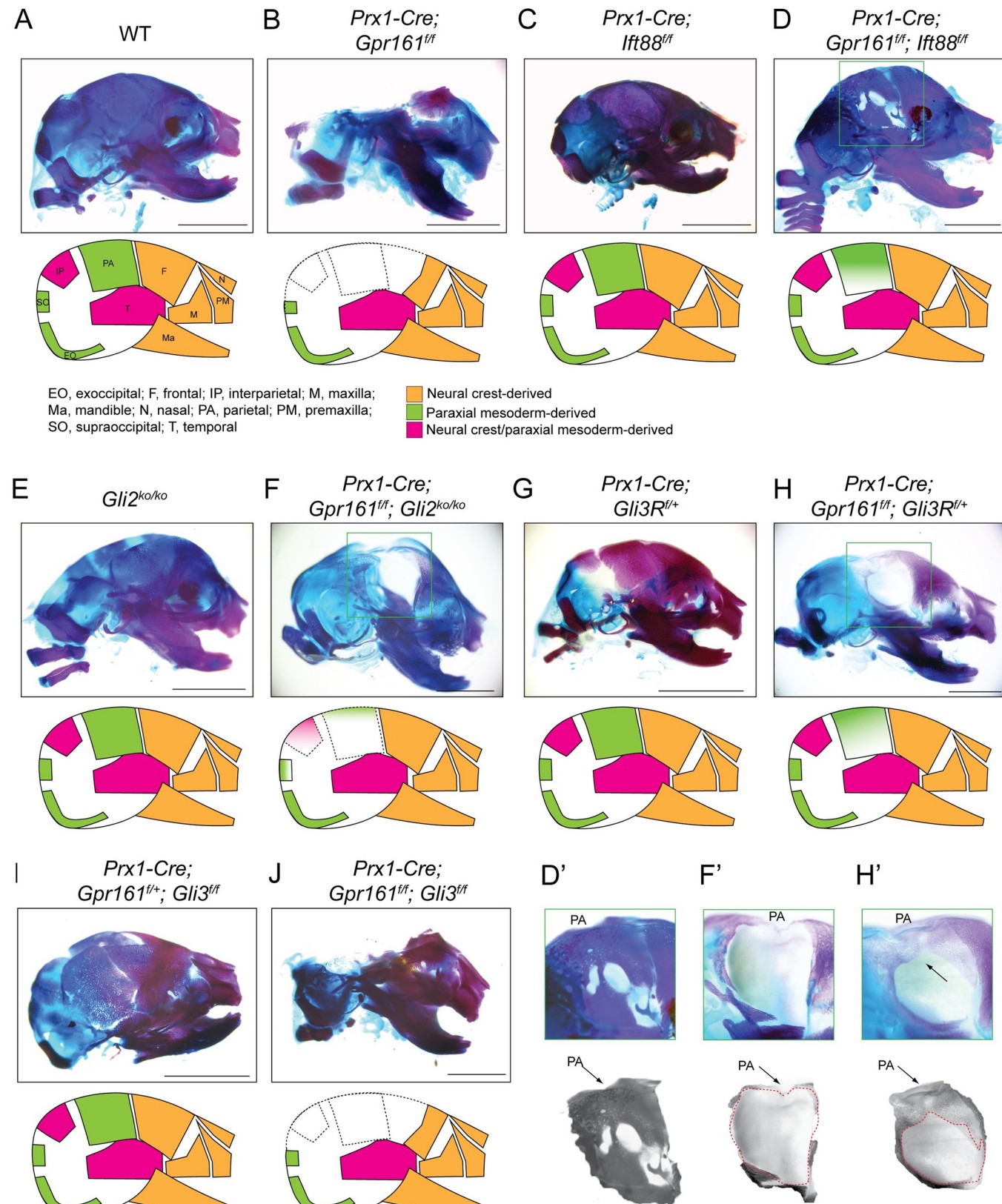

EO, exoccipital; F, frontal; IP, interparietal; M, maxilla;
Ma, mandible; N, nasal; PA, parietal; PM, premaxilla;
SO, supraoccipital; T, temporal

Neural crest-derived
Paraxial mesoderm-derived
Neural crest/paraxial mesoderm-derived

**Fig 5. GLI2/GLI3R dependance on calvaria mineralization in *Prx1-Cre; Gpr161^{f/f}*. (A-J)** Alcian blue (unmineralized cartilage) and alizarin red (mineralized cartilage and bone) staining of cranium in E18.5 embryos. The number of calvaria analyzed (n) for the genotypes shown are as follows: (A) *Prx1-Cre; Gpr161^{f/+}* (WT, n = 5), (B) *Prx1-Cre; Gpr161^{f/f}* (n = 9), (C) *Prx1-Cre; Ift88^{f/f}* (n = 5), (D) *Prx1-Cre; Ift88^{f/f}; Gpr161^{f/f}* (n = 6), (E) *Gli2* ko (n = 4), (F) *Prx1-Cre; Gpr161^{f/f}, Gli2* ko (n = 3), (G) *Prx1-Cre; Gli3R^{f/+}* (n = 3), (H) *Prx1-Cre; Gpr161^{f/f}; Gli3R^{f/+}* (n = 3), (I) *Prx1-Cre; Gpr161^{f/+}; Gli3^{f/f}* (n = 3), (J) *Prx1-Cre; Gpr161^{f/f}, Gli3^{f/f}* (n = 3). D', F' and H' show expanded views of parietal. Representative images from each genotype are shown. The relationships of the cranial bones to neural crest/paraxial mesoderm shown in the schematic is based on [55,56]. Abbreviations: EO, exoccipital; F, frontal; IP, interparietal; M, maxilla; Ma, mandible; N, nasal; PA, parietal; PM, premaxilla; SO, supraoccipital; T, temporal. Scale, 2 mm (A-J).

in radius and ulna but was reduced to absent in humerus, compared to complete lack of mineralization in forearm long bones in *Prx1-Cre; Gpr161^{f/f}* (**Figs 7A–7C, 7E and S5**). *Prx1-Cre; Gpr161^{mut1/f}* embryos also showed a lack of mineralization and osteogenesis in only parietal bones with normal mineralization in other calvarial bones (**Fig 7D**). The forelimb autopod of *Prx1-Cre; Gpr161^{mut1/f}* was polysyndactylous (**S3 Fig**).

Overall, the phenotypes of *Prx1-Cre; Gpr161^{mut1/f}* mostly resembled *Prx1-Cre; Gpr161^{f/f}; Gli2* ko embryos; correspondingly, the regions that manifested lack of mineralization phenotypes in *Prx1-Cre; Gpr161^{mut1/f}* embryos, such as humerus and medial parietal, were rescued by concomitant *Gli3R* expression in *Prx1-Cre; Gpr161^{f/f}* embryos. Thus, lack of GPR161 from cilia phenocopied GPR161 loss of function mediated predominantly from GLI3R loss but not from the generation of GLI2A.

## Discussion

How signaling emanating from cilia mediates mesoscale tissue organization is not well understood. We suggest three distinct modes by which the cilia generated GLI effectors regulate morpho-phenotypic outcomes arising from HH pathway derepression (**Fig 8A**) [6,41]. These modes of regulation include:

a. ratio sensing between GLI-A and GLI-R levels,

b. detecting thresholds of GLI-A levels or

c. detecting thresholds of GLI-R levels.

In a *ratio sensing* tissue region, the relative levels of GLI-A vs GLI-R matter for manifestation of the derepression phenotype. Thus, if a phenotype is caused by a decrease in GLI-R and increase of GLI-A, loss of GLI2 or introduction of GLI3R would rescue the phenotype. The manifestation of the HH pathway derepression phenotype in this region should not require SMO, as the lack of GLI-R likely results in generation of sufficient amount of GLI-A. We find that the regions regulated in a ratio sensing mode are the following: most of the dorsal neural tube, most of radius and ulna, and most of the calvarium (except medial parietal bone) (**Fig 8B–8D**).

In a *threshold activator* tissue region, only the levels of GLI-A matter for manifestation of the derepression phenotype. Here, loss of GLI2 but not *Gli3R* expression would rescue the phenotype. As GLI-A primarily manifests the phenotypes, SMO should be required in some regions depending on the amount of GLI-A required. One of regions that is regulated in the threshold activator mode includes the ventral most neural tube progenitors in the floor plate region (**Fig 8B**). GLI2 activation is also a key driver of SHH subset of medulloblastomas, which originate from granule cell progenitors [42].

In a *threshold repressor* tissue region for manifestation of the derepression phenotype, only the relative levels of GLI-R matter. Here, only an increase in GLI3R levels but not GLI2 loss would rescue the phenotype. As GLI-R lack solely manifests the phenotype in this region, SMO should not be required. The regions that are regulated in a repressor mode are: humerus (**Fig 8C**) and medial parietal bone (**Fig 8D**).

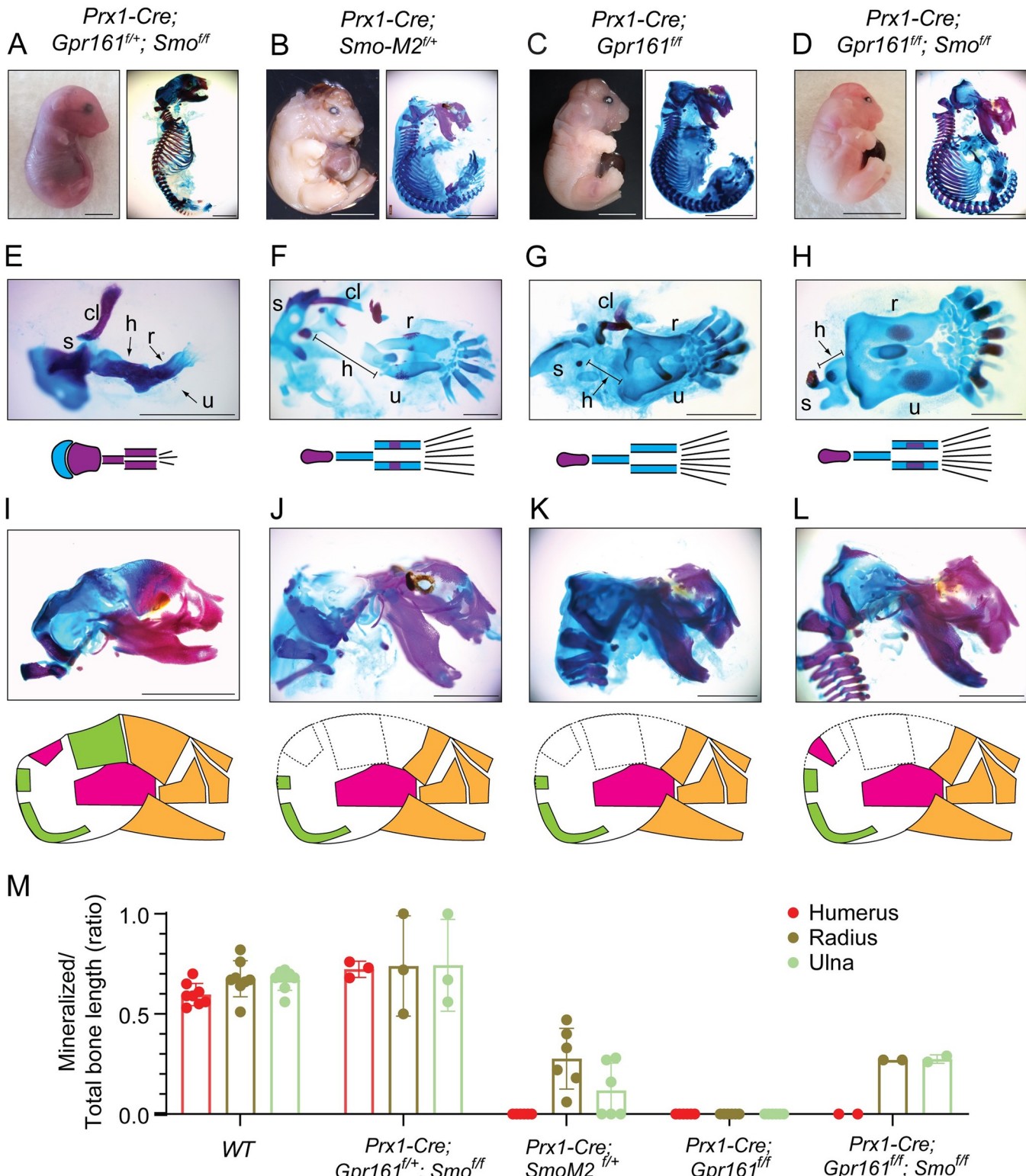

**Fig 6. Forelimb long bone and calvarial mineralization in *Prx1-Cre; Gpr161^{f/f}* was mostly SMO independent.** (A-L) Whole embryo gross features (top left, lateral view), and Alcian blue (unmineralized cartilage) and alizarin red (mineralized cartilage and bone) staining of full skeleton (top right, lateral view) (A-D), forelimbs (E-H) and cranium (I-L) in E18.5 embryos. Genotypes analyzed are as follows: (A, E, I) *Prx1-Cre; Gpr161^{f/+}; Smo^{f/f}* (n = 6), (B, F, J) *Prx1-Cre; SmoM2^{f/+}* (n = 4), (C, G, K) *Prx1-Cre; Gpr161^{f/f}* (n = 9), (D, H, L) *Prx1-Cre; Gpr161^{f/f}; Smo^{f/f}* (n = 2). **(M)** Quantification of mineralized to total bone primordium length ratios shown. Each data point represents individual bone primordia quantified. Abbreviations: clavicle, cl, clavicle; scapula, s, scapula; h,

humerus; r, radius; u, ulna; Cranial bones are shown as in Fig 5A. Scale, 5 mm (A-D), 2 mm (E-L). See also S4 Fig for quantification of forearm long bone primordium lengths and S3 Fig showing autopods.

## Dual regulation of HH pathway derepression by *Gli3R* loss and GLI2A generation

We find that most HH pathway derepression phenotypes manifest in a range where either a lack of GLI3R or presence of GLI2A can rescue the phenotypes (**Fig 8B–8D**). Previously, using *Gli3* ko and *Gli3R* expression, the intermediate neural tube was thought to be solely GliR-dependent [27]. Our data shows that ventral progenitor marker expression in most of the dorsolateral neural tube except ventral most floorplate and p3 region is regulated by either GLI3R expression or from GLI2 loss. Such dual regulation would be expected, based on the scenario that the GLI-A and GLI-R likely regulate GLI binding sites reciprocally. However, such regulation also brings up interesting consequences for cells that are thought to be fully HH dependent. For e.g., the expression of FOXA2 in the floor plate and NKX2.2 in the p3 domain is fully SHH and GLI2 dependent. However, both FOXA2 and NKX2.2 expressing neuroprogenitors are repressed beyond the floor plate and p3 domains of the neural tube by either *Gli3R* expression or from GLI2 loss (**Fig 8B**). Thus, ectopic expression of FOXA2 or NKX2.2 can be regulated in distinct modes in different regions of the neural tube: fully threshold activator mode in the floor plate and ratio sensing mode in rest of the neural tube. Such mode of dual regulation by either *Gli3R* expression or from GLI2 loss also extends for rescue of mineralization in radius or ulna and in most of the calvarium in *Prx1-Cre; Gpr161^{f/f}*.

## Requirement for threshold activation in certain HH pathway derepression phenotypes

Correspondingly, there are some derepression phenotypes that require a boost in expression of GLI2A. It is rather counterintuitive that why a lack of GLI-R formation by cilia ultimately manifests in phenotypes that are regulated by GLI2A formation. The derepression phenotypes that require very high GLI2A thresholds are the ones that are primarily dependent on GLI2, i.e., that are only rescued from concomitant GLI2 loss but not from *Gli3R* expression. For example, the ventral most expression of floor plate progenitors in neural tube in *Gpr161* ko are restored upon concomitant lack of GLI2. If excess GLI2A is needed in a SMO-dependent manner, SMO loss would rescue some of the phenotypes; for example, the decreased overall expression levels of FoxA2 in neural tube in *Gpr161; Smo* double ko [18]. However, floor plate progenitors are expressed throughout the dorso-ventral *Ankmy2* ko neural tube even in the absence of SMO [17]. In this case, a complete loss of GLI2R and GLI3R probably ensures adequate production of both GLI2A and GLI3A independent of SMO in facilitating the *FoxA2* expression.

## Tissue specific GLI-R thresholds in regulating HH repression

Our results point to the role of tissue specific GLI-R thresholds in regulating HH repression during morphogenesis (**Fig 8E**). In this scenario, some tissues such as humerus, medial parietal bone, and frontonasal processes [21] requires high level of GLI3R to be expressed for preventing derepression phenotypes in these regions. Lack of cilia in the face causes widening of the face by increasing distance between frontonasal processes that phenocopies *Gli2; Gli3* double ko or *Gpr161* conditional ko [21,43]. The lack of a *Gli2R* allele prevents from assessing the precise function of Gli2R, but *Gli3R* expression rescues this phenotype, suggesting predominant role of GLI-R [43]. The high GLI3R levels could prevent expression of unknown gene targets

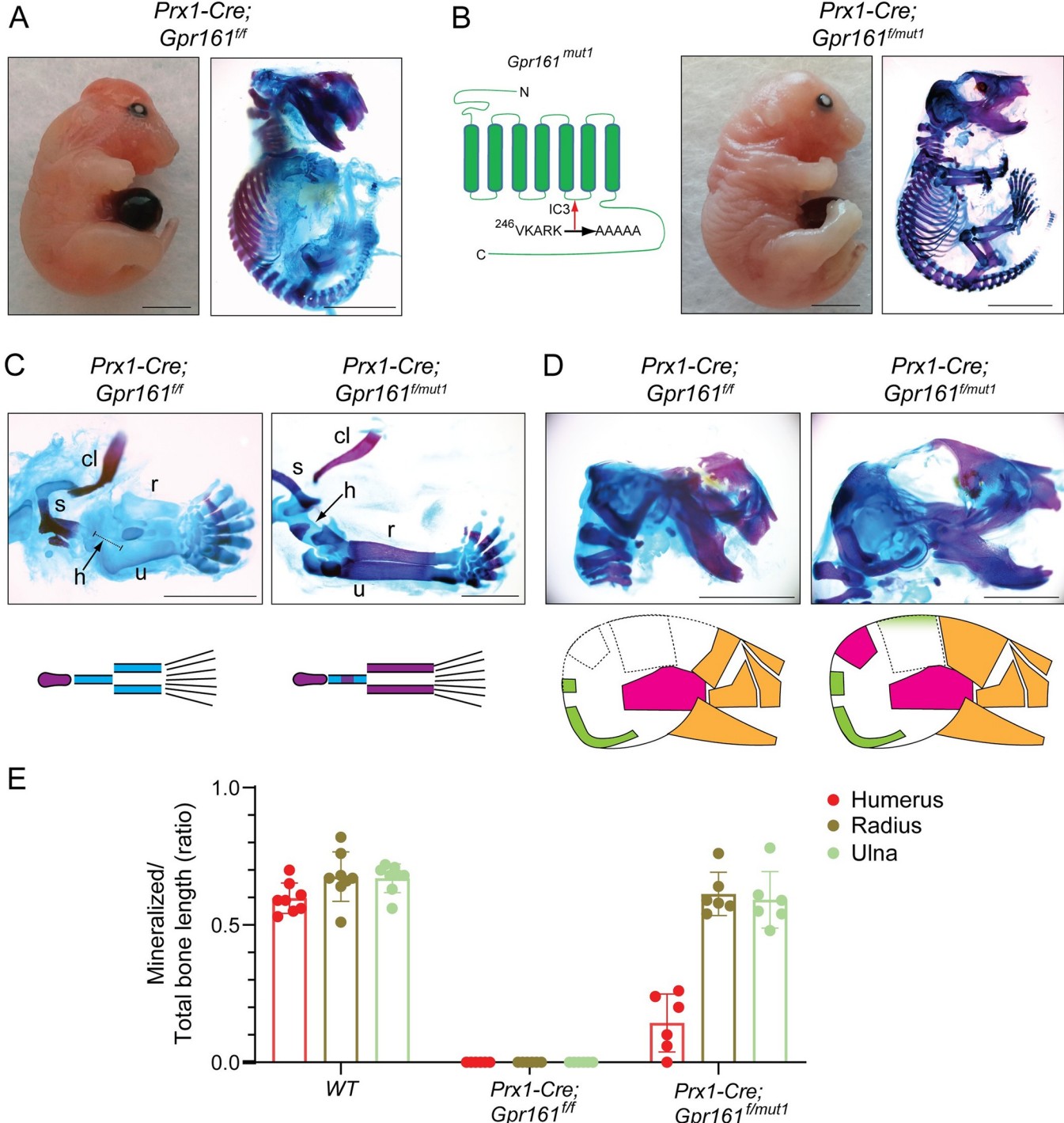

**Fig 7. Lack of GPR161 from cilia manifested phenotypes predominantly mediated from GLI3R loss. (A-D)** Whole embryo gross features (top left, lateral view), and Alcian blue (unmineralized cartilage) and alizarin red (mineralized cartilage and bone) staining of full skeleton (top right, lateral view) (A-B), forelimbs (C) and cranium (D) in E18.5 embryos. Genotypes analyzed are as follows: (A, C, D) *Prx1-Cre; Gpr161^{f/f}* (n = 15) and (B, C, D) *Prx1-Cre; Gpr161^{f/mut1}* (n = 5). Cartoon representing the VKARK>AAAAA *mut1* mutation in the third intracellular loop of mouse GPR161. **(E)** Quantification of mineralized to total bone primordium length ratios shown. Each data point represents individual bone primordia quantified. Abbreviations: clavicle, cl, clavicle; scapula, s, scapula; h, humerus; r, radius; u, ulna; Cranial bones are shown as in Fig 5A. Scale, 5 mm (AB), 2 mm (C-D). See also S5 Fig for quantification of forearm long bone primordium lengths and S3 Fig showing autopods.

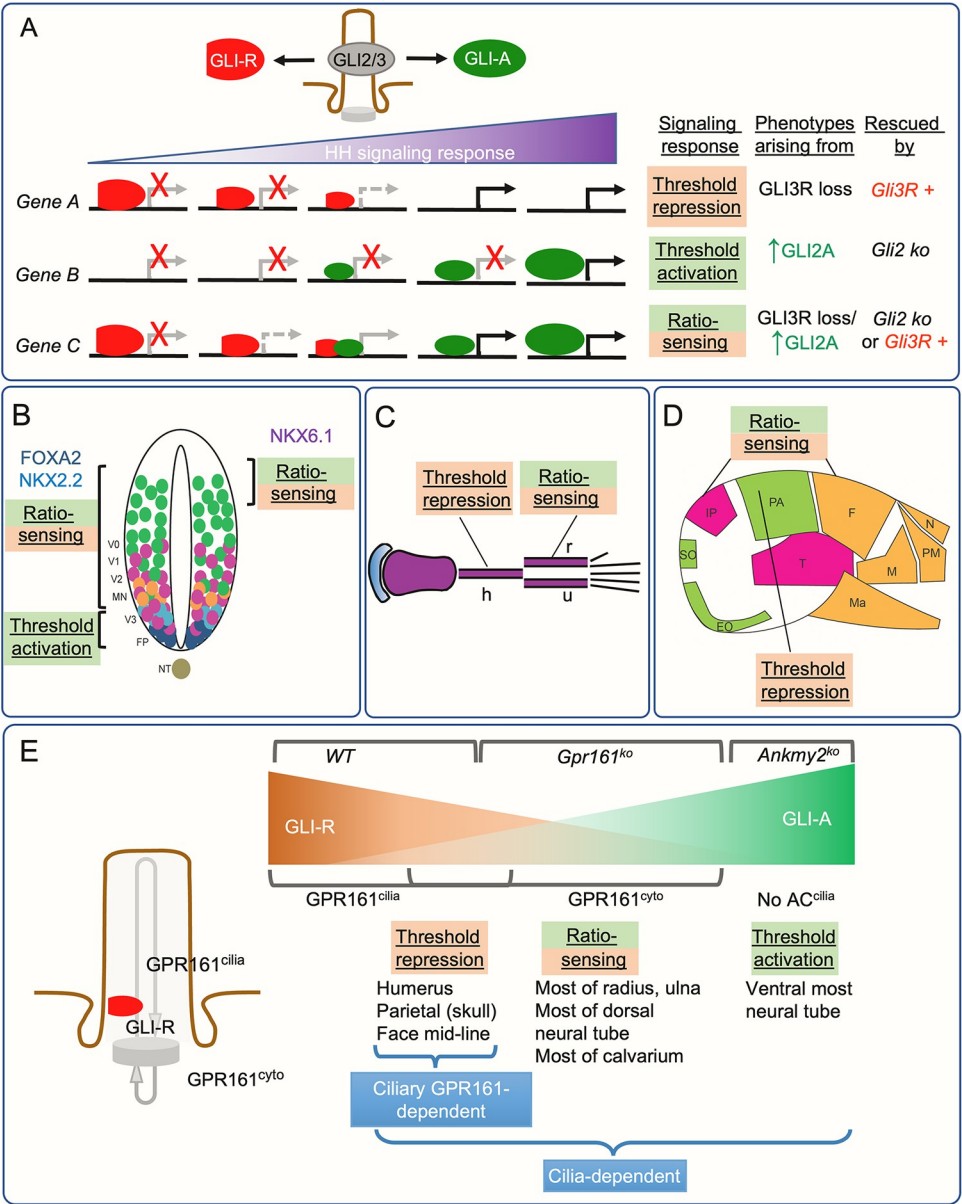

**Fig 8. Distinct modes by which the cilia generated GLI effectors regulate morpho-phenotypic outcomes arising from HH pathway derepression. (A)** HH signaling outputs are regulated by bi-functional GLI transcriptional factors that can function as both activators and repressors. The distinct modes include (a) ratio sensing between GLI-A and GLI-R levels, (b) detecting thresholds of GLI-A or (c) detecting thresholds of GLI-R levels. **(B-D)** Distinct modes of regulation shown in HH pathway derepression phenotypes in neural tube (B), forearm long bones (C), and cranium (D) as described in Discussion. The neuronal subtype domains are depicted as in Fig 2C. **(E)** Role of tissue specific GLI-R thresholds in regulating hedgehog repression during morphogenesis established by ciliary and extraciliary pools of GPR161 and ciliary AC trafficking by ANKMY2. Abbreviations: h, humerus; r, radius; u, ulna; EO, exoccipital; F, frontal; IP, interparietal; M, maxilla; Ma, mandible; N, nasal; NT, notochord; P, parietal; PM, premaxilla; SO, supraoccipital; T, temporal.

that would manifest in these phenotypes. Lack of GLI2A cannot rescue these phenotypes as these gene targets are primarily prevented from baseline or leaky expression by GLI3R. Thus, GLI3R is preventing baseline or leaky expression likely by transcriptional activators other than GLI2A in these contexts.

## Cilia generated signaling in GLI-R threshold generation

We recently showed that GPR161 can form GLI-R by localizing both inside cilia and in peri-ciliary regions [21]. In addition, disruption of GPR161 trafficking to cilia [21] or from loss of GPR161 PKA anchoring activity [44], also results in accumulation of GLI2 in ciliary tips of resting cells. Thus, the ciliary pools of GPR161 accentuate GLI3R generation and prevent GLI2 accumulation in ciliary tips. We now show that cilium-specific signaling generated by GPR161 prevent GLI repression threshold dependent phenotypes (**Fig 8E**). In this case, lack of GPR161 from cilia recapitulates phenotypes ensuing mostly from GLI3R loss alone in *Gpr161* deletion background. Rather, GPR161 lack from cilia resembles concomitant deletion of *Gpr161* and *Gli2*, where GLI3R level decrease in *Gpr161* ko is not restored. Such correspondence in pheno-types are seen for lack of mineralization phenotypes in *Prx1-Cre; Gpr161^{mut1/f}* humerus and parietal regions, both of which are not rescued in *Prx1-Cre; Gpr161^{f/f}; Gli2* ko. Overall, these results suggest that GPR161 levels inside cilia are critical in establishing high GLI3R levels, and tissues that require high GLI3R thresholds to suppress leaky expression of targets are affected from GPR161 absence from cilia.

## Conclusion and future directions

Why are some of the derepression phenotypes rescued selectively by GLI2 loss or *Gli3R* expres-sion? Why does tissues require thresholds of GLI-R in repression of phenotypes? Are these modes of regulation relevant during hedgehog pathway mediated tissue regeneration, such as in muscle-resident fibro/adipogenic progenitors [45]? Whereas GLI family members bind to the same consensus DNA sequences, the regulation of cis-regulatory modules of targets are complex, context-dependent, and likely regulatable by co-activators and repressors [46]. It has been shown that GLI3 can couple with HAND2 in coordinating skeletal gene regulatory net-works in the mandibular region [47]. The mandibular prominence region showing such regu-lation is outside of the highest SHH ligand concentration. Similarly, our results suggest GLI2-independent but co-transactivator mediated responses in manifesting derepression phe-notypes in tissues regulated solely by GLI-R thresholds. In contrast, GLI3R has been proposed to be inert in repression in the limb bud prior to SHH signaling [48]. One way to reconcile these differences between tissues could be from context dependent GLI3R function in exhibi-tion of derepression phenotypes. Collectively, our results establish a rational framework for different modes of GLI-A/R regulation and GLI-R thresholds regulated by cilia-generated sig-naling underlying HH repression in diverse tissues.

## Materials and methods

### Ethics statement

All the animals in the study were handled according to protocols approved by the UT South-western Institutional Animal Care and Use Committee, and the mouse colonies were main-tained in a barrier facility at UT Southwestern, in agreement with the State of Texas legal and ethical standards of animal care.

### Mouse strains

All mice were housed at the Animal Resource Center of the University of Texas Southwestern (UTSW) Medical Center. All protocols were approved by the UTSW Institutional Animal Care and Use Committee. Mice were housed in standard cages that contained three to five mice per cage, with water and standard diet *ad libitum* and a 12 h light/dark cycle. Both male and female mice were analyzed in all experiments. *Prx1-Cre* mice (Stock No. 005584; MGI:

2450929) [49] were obtained from Jackson Laboratory (Bar Harbor, ME) [50]. The *Gpr161* knockout and conditional allele targeting the third exon crossed has been described before [19]. The *Gpr161* knockout and conditional lines are already indexed in MGI as *Gpr161^tm1.1Smuk^*, MGI: 6357708; *Gpr161^tm1.2Smuk^*, MGI: 6357710). Double knockout analysis was performed using *Gli2^tm1Alj^* (ko) allele [51]. *Prx1-Cre* ([49]; Jax strain No: 005584) was crossed with the *Gpr161^f/f^*. The *Gli2 ko* and *Gpr161* floxed alleles were linked through genetic recombination by breeding *Gpr161^f/f^* with *Gli2^ko/+^* animals. Crossing with *CAG-Cre* recombinase line [52], in which Cre is expressed ubiquitously, generated the linked *Gpr161; Gli2* double knockout allele. Yolk sac DNA was used for genotyping embryos. ES cells for *Gli3^Δ701C^* [20] were injected by the UT Southwestern transgenic core into C57BL/6N blastocysts and selected for germline transmission among chimeras and backcrossed into C57BL/6J background. Heterozygotes of the ubiquitously recombined *Gli3^Δ701C^* allele are embryonic lethal [20], suggesting that *Gli3^Δ701^* allele produces a much more potent form of GLI3R than the widely used *Gli3^Δ699^* allele yielding viable heterozygotes [30]. Therefore, mice with *Gli3^Δ701C/Δ701C^* alleles were crossed with *CAG-Cre* [52] or *Prx1-Cre* [49] along with *Ankmy2* ko, *Gpr161* ko or *Gpr161^f^* alleles to generate *Gli3R* containing genotypes in *Ankmy2* ko, *Gpr161* ko or conditional ko background, respectively. Noon of the day on which a vaginal plug was found was considered E0.5.

## Mouse genotyping

To genotype *Ankmy2* mice, following primers were used: 3F (5′-CTG TCT CCA TAT TCA CAC ATT GAA TAG C-3′), 4R (5′-GCT GCA TGC ATC AAA GGA GTC ATT CC-3′) and 5R (5′-CAA CGG GTT CTT CTG TTA GTC C-3′). For ko allele: 3F, 4R and 5R gave 508 bp and 316 bp for wild type and knockout bands, respectively [17]. Genotyping of *Gpr161 mut1* alleles were performed using primers in the intron 3–4 (5′ CAGAAAGCAACAGCA AAGCA) and intron 4–5 (5′-ACCCTGACACTGCCCTTAGC). The PCR product of wild type and *mut1* allele bands was 927 bp, but only the PCR product from the *mut1* allele was digested into 400 and 527 bp products with NotI. Genotyping of *Gpr161* knockout or floxed alleles were performed using primers in the deleted 4^th^ exon (5′-CAAGATGGATTCGCAGTAGC TTGG), flanking the 3′ end of the deleted exon (5′ ATGGGGTACACCATTGGATACAGG), and in the Neo cassette (5′-CAACGGGTTCTTCTGTTAGTCC). Wild type, floxed and knockout bands were 816, 965, and 485 bp, respectively [19]. *Cre* allele was genotyped with Cre-F (5′-AAT GCT GTC ACT TGG TCG TGG C-3′) and Cre-R (5′-GAA AAT GCT TCT GTC CGT TTG C-3′) primers (100 bp amplicon). To genotype *GLI2* mice, GLI2 sense (5′-AAA CAA AGC TCC TGT ACA CG-3′), GLI2 antisense (5′-CAC CCC AAA GCA TGT GTT TT-3′) and pPNT (5′-ATG CCT GCT CTT TAC TGA AG-3′) primers were used. Wild type and knockout bands were 300 bp and 600 bp, respectively. Genotyping of *Gli3^Δ701C^* allele was done with the following primers: BW892F, 5′-AATGGAATGTTTCCAAGACTG-3′, and BW892R, 5′-ATAAAACCAAGGGTTCCAGATC-3′, with wild-type and mutant bands being 180 bp and 250 bp, respectively [20]. *Cre*-mediated recombination was confirmed using BW791, 5′-GACCTCATCTTTAGCTTTGCC-3′, and BW1020R, 5′-CAAGGGTTCCAGA TCTGGATC-3′, the recombined allele band being 230 bp.

## Tissue processing, immunostaining, and microscopy

Mouse embryos fixed in 4% PFA overnight at 4°C and processed for cryosectioning. For cryosectioning, the embryos were incubated in 30% sucrose at 4°C until they were submerged in the solution. Embryos were mounted with OCT compound. Embryos in OCT were cut into 15 μm frozen sections. The sections were incubated in PBS for 15 min to dissolve away the

OCT. Sections were then blocked using blocking buffer (1% normal donkey serum [Jackson immunoResearch, West Grove, PA] in PBS) for 1 h at room temperature. Sections were incubated with primary antibodies against the following antigens; overnight at 4°C: FOXA2 (1:1000, ab108422; Abcam), NKX2.2 (1:10, 74.5A5-s; DSHB), OLIG2 (1:500, MABN50; Millipore), NKX6.1 (1:100, F55A10-s; DSHB), PAX6 (1:2000, 901301; Biolegend). After three PBS washes, the sections were incubated in secondary antibodies (Alexa Fluor 488-, 594-, 647- conjugated secondary antibodies, 1:500; Life Technologies, Carlsbad, CA or Jackson ImmunoResearch) for 1 h at room temperature. Cell nuclei were stained with DAPI. Slides were mounted with Fluoromount-G (0100–01; Southern Biotech) and images were acquired with a Zeiss AxioImager.Z1 microscope.

## Skeletal staining

Skeletal preparations were made by a slight modification of the Alcian blue/alizarin red staining procedure described by [53]. Specimens were fixed in 99% ethanol for 24 h (embryos older than day E15 were first deskinned and eviscerated), and then kept in acetone for another 24 h. Incubation in staining solution (1 volume of 0.3% Alcian blue in 70% ethanol, 1 volume of 0.1% alizarin red S in 96% ethanol, 1 volume of absolute acetic acid, and 17 volumes of 70% ethanol) was performed for 2–3 days at 37°C. Samples were rinsed in water and kept in 1% potassium hydroxide/20% glycerol at 37°C overnight, with additional incubation at room temperature until complete clearing. For long-term storage, specimens were transferred into 50%, 80% and finally 100% glycerol. Images were acquired using a Leica stereomicroscope (M165 C) with digital camera (DFC500).

## Quantification of phenotypes

Quantification of FOXA2, NKX2.1 and NKX6.1 expression to that of the neural tube was done from sections at thoracic and lumbar region utilizing the line tool in Image J [21]. Measurement of the entire length of the bone primordium was done utilizing the line tool in Image J [54]. All numerical data is in a spreadsheet form as supporting information, please see S1 Data.

## Supporting information

**S1 Fig.** *Gli3R* **expression partially restored HH signaling dependent progenitors in** *Ankmy2* **ko hindbrain.** Top, panels show bright-field images of wildtype (*wt*), *Ankmy2 ko*, *Gli3R/+*, *Ankmy2 ko; Gli3R/+*, *Gli2 ko*, *Ankmy2; Gli2* double *ko*, and whole-mount embryos at E9.25 of the following genotypes: wild-type (n = 5), *Ankmy2* ko (n = 5), *Gli3R/+* (n = 4), *Ankmy2* ko; *Gli3R/+* (n = 4), *Gli2* ko (n = 3), *Ankmy2* ko; *Gli2* ko (n = 3). Exencephaly (marked by asterisk) persists in *Ankmy2* ko and double mutants. Scale: 500 μm. Bottom, panels show hindbrain neural tube horizontal sections immunostained using designated markers. All images are counterstained with DAPI. Scale: 100 μm.
(TIF)

**S2 Fig. GLI2/GLI3R dependance on forelimb long bone mineralization in** *Prx1-Cre; Gpr161f/f*. Quantification of forearm long bone primordia lengths in Fig 4 shown. Each data point represents individual bone primordia quantified. Alcian blue (unmineralized cartilage) and alizarin red (mineralized cartilage and bone) staining of E18.5 embryos shown below. Scale, 5 mm.
(TIF)

**S3 Fig. GLI2/GLI3R dependance on autopod patterning in** *Prx1-Cre; Gpr161f/f* **autopods.** Alcian blue (unmineralized cartilage) and alizarin red (mineralized cartilage and bone)

staining of E18.5 forelimb autopods. Duplicated or unassigned digits are marked by asterisks, whereas replicated carpals/metacarpals are hyphenated with likely identities. Quantification of whole digit numbers (without including duplicated phalanges or thin elongations) are shown below right. Each shape represents an individual autopod quantified and total number of autopods quantified (n) are shown on top of the graph. Abbreviations: c, Capitate; cc, Central carpal; h, Hamate; mc, Metacarpal; s, Scaphoid; td, Trapezoid; tm, Trapezium; t, Triquetral. Scale, 1 mm.
(TIF)

**S4 Fig. Forelimb long bone and calvarial mineralization in *Prx1-Cre; Gpr161^{f/f}* is mostly SMO independent.** Quantification of forearm long bone primordia lengths in Fig 6 shown. Each data point represents individual bone primordia quantified.
(TIF)

**S5 Fig. Lack of GPR161 from cilia manifested phenotypes predominantly mediated from GLI3R loss.** Quantification of forearm long bone primordia lengths in Fig 7 shown. Each data point represents individual bone primordia quantified.
(TIF)

**S1 Data. Numerical data in figures and supplemental figures.** All numerical data in figures and supplemental figures are provided in this spreadsheet.
(XLSX)

## Acknowledgments

We thank the transgenic core, molecular pathology, and mouse animal care facility in UT Southwestern. We thank John Shelton for histopathology core support. Monoclonal antibodies developed by O.D. Madsen were obtained from the Developmental Studies Hybridoma Bank developed under the auspices of the NICHD and maintained by the Department of Biological Sciences, the University of Iowa, Iowa City, IA, USA.

## Author Contributions

**Conceptualization:** Sun-Hee Hwang, Saikat Mukhopadhyay.

**Data curation:** Sun-Hee Hwang.

**Formal analysis:** Saikat Mukhopadhyay.

**Funding acquisition:** Baolin Wang, Saikat Mukhopadhyay.

**Investigation:** Sun-Hee Hwang, Kevin Andrew White, Bandarigoda Nipunika Somatilaka.

**Methodology:** Sun-Hee Hwang, Kevin Andrew White, Bandarigoda Nipunika Somatilaka, Baolin Wang.

**Project administration:** Saikat Mukhopadhyay.

**Supervision:** Saikat Mukhopadhyay.

**Validation:** Sun-Hee Hwang, Kevin Andrew White, Bandarigoda Nipunika Somatilaka.

**Visualization:** Sun-Hee Hwang, Kevin Andrew White.

**Writing – original draft:** Saikat Mukhopadhyay.

**Writing – review & editing:** Sun-Hee Hwang, Kevin Andrew White, Bandarigoda Nipunika Somatilaka, Baolin Wang, Saikat Mukhopadhyay.

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
