## [Decision Letter · Decision Letter 0]

22 Aug 2023

Dear Dr Mukhopadhyay,

Thank you very much for submitting your Research Article entitled 'Context-dependent ciliary regulation of hedgehog pathway repression in tissue morphogenesis' to PLOS Genetics.

The manuscript was fully evaluated at the editorial level and by independent peer reviewers. The reviewers appreciated the attention to an important topic but identified some concerns that we ask you address in a revised manuscript.  Reviewer 1 suggested analyzing GLI-R/R homozygous embryos. Please address this concern with text edits where appropriate.

We therefore ask you to modify the manuscript according to the review recommendations. Your revisions should address the specific points made by each reviewer.

Yours sincerely,

Gregory J. Pazour

Guest Editor

PLOS Genetics

Gregory Barsh

Editor-in-Chief

PLOS Genetics

Reviewer's Responses to Questions

**Comments to the Authors:**

Reviewer #1: In this study, Hwang and colleagues examined the relative contributions of GLI-Activator (GLI-A) and GLI-Repressor (GLI-R) to determine how they regulate development in three different tissues. Using mutations that limit GLI function, they then examine the consequences of restoring GLI-R or GLI-A to the observed phenotypes of the neural tube, long bones and calvaria. This study is asking an important longstanding question about how Hedgehog signaling regulates different tissues in different ways. As aspects of both the Ankmy2 and Gpr161 phenotype characterizations are partially redundant with previous studies, the data is not all novel but there is real value in aggregating this into one focused study. The data is of high technical quality and I am overall enthusiastic about it although I do have some concerns with the study in its present form. First, while the neural tube provides a well-defined system for examining GLI readouts, interpreting the calvaria and long bone phenotypes is trickier because bone mineralization is a rather vague phenotype and it is unclear when GLI proteins regulate these tissues and what specific cell types are impacted. In addition, I am concerned about the reliance on GLI3-R/+ heterozygous embryos instead of GLI-R/R homozygous compound crosses that would provide more definitive information about the role of GLI repression. Despite these concerns and a few additional minor points all outlined below, this study provides a valuable framework for understanding how GLI-A and GLI-R interact with distinct mechanisms in different tissues.

Major concerns:

1. The study relies on GLI3-R/+ heterozygous embryos instead of GLI-R/R homozygous embryos for interpreting phenotypes caused by GLI repression. But as described by the Wang lab, heterozygous GL3-R/+ embryos do not fully copy the Shh-/- embryos like the homozygous alleles, indicating that both copies of GLI3-R are required for full GLI repression. While this does not need to be examined for all mutants/condition, it is important to determine if the dorsal NKX2.2 in the Ankmy2-/-;GLI3R/+ neural tube goes away completely in Ankmy2-/-;GLI3R/R as predicted by the model.

2. The lack of bone mineralization used as a phenotype for GLI-A and GLI-R requirements is problematic for two reasons. First, this is an indirect readout and it is unclear what is the affected cell type (osteoblast specification? chondrocyte morphology and maturation?). Second it is unclear when GLI signaling regulates bone development. Does this occur sequentially in the limb bud and then in the bone, a complicating factor in considering ratio sensing? Are the missing skull bones a consequence of direct or indirect Hedgehog signaling? And how are crest and non-neural crest derived tissues affected at the same time? These caveats should be acknowledged and discussed both within the results and the discussion.

Minor issues:

3. Figures 2 and 3. In contrast to other markers, it isn’t easy to see the signal in all the NKX2.2 panels (both in Figs. 2 and 3) and this is critical for interpreting GLI-R readouts. Higher resolution images, perhaps without DAPI overlay might help resolve this.

4. Figure S1. Does not indicate the number of embryos examined per genotype. This should be added to the legend.

5. Typo (stray \\) on line 197

6. Figure 4A,B and line 252. It is stated that the length of the humerus is reduced in Prx1-Cre;Gpr161f/f forelimbs but it isn’t possible to clearly see the humerus in the image provided and it isn’t clear to me from looking at 4B that it is even present. Could this be replaced with a clearer image?

7. Figure 4E. The humerus does not see completely mineralized as indicated on the schematic lying directly below, which seems to be a mistake based on their stated model.

8. Figure S3. State the N’s for each specific cross in the figure legend and also to what degree the phenotypes are penetrant. Since there is highly likely to be variation, it would be nice to quantify each of these mutant conditions to show their penetrance.

9. Figure 5 legend. Please add N’s for each specific genotype.

Reviewer #2: This manuscript from Mukhopadhyay and colleagues focuses on the GLI2 and GLI3 proteins, which serve as the major transcriptional effectors of the Hedgehog (Hh) pathway. GLI2 and GLI3 are bifunctional proteins that can exist in activator or repressor forms; while cilia are needed for both GLI repressor and activator formation, the degree to which ciliary Hh signaling (and cilium-related phenotypes) rely on repressors, activators, or both remains incompletely understood.

To address these questions, the authors employ genetic and embryological approaches to investigate the roles of GLI2 and GLI3 during morphogenetic Hh signaling in two well-established experimental models -- neural tube and skeletal development. They took advantage of two genetic manipulations that derepress ciliary Hh signaling to different extents: 1) loss of Ankmy2, which serves to traffic adenylyl cyclases to cilia; 2) loss or ciliary delocalization of Gpr161, a constitutively active GPCR that helps to set baseline PKA activity in cilia. To study the role of GLI activators and repressors, the authors combined the above genetic manipulations with either: 1) enforced expression of GLI3 repressor (GLI3R); 2) loss of Gli2, the major transcriptional activator of the Hh pathway. The authors go on to conduct a careful and comprehensive epistasis analysis, including appropriate controls and quantifications.

The key finding is that during neural tube and skeletal development, some Gpr161 or Ankmy2 phenotypes are rescued by either GLI3R expression or Gli2 knockout, whereas others can only be rescued by one or the other. Based on this finding, the authors articulate a model, building on the one they proposed in their 2021 eLife paper, that three distinct modes of GLI target gene regulation (ratio sensing, GLI3R threshold, and GLI2A threshold) carry out the transcriptional effects of Hh signaling.

Overall this is a high-quality and thorough genetic analysis that is suitable for publication in PLOS Genetics, pending the following minor revisions to the text and figures:

1. The authors assume that GLI2 functions primarily as an activator in their experiments. Is it possible that GLI2 repressor plays any roles? A limitation of Gli2 knockout is that it removes both GLI2R and GLI2A. This is a reasonable assumption because GLI2 undergoes far less proteolytic processing to a repressor than does GLI3. Nevertheless, given that the authors made a point of emphasizing how a Gli2 or Gli3 knockout cannot distinguish between activator and repressor roles (p. 4-5), and given that the authors don’t have a way to explicitly test GLI2R like they do with the GLI3(delta701) allele, it’s worth acknowledging this possibility as a caveat of the GLI2 knockout studies.

2. It is interesting that the Ankmy2 exencephaly phenotype is not overcome by either GLI3R expression or Gli2 knockout, and therefore doesn’t appear to fall into one of the 3 modes of GLI regulation in the authors’ model. Can the authors offer any potential explanation for this result?

3. I am confused about blue color scheme in the cartoons for the calvarium studies (i.e., Fig 5D) and what it’s supposed to represent. Please clarify this in the figure legend.

4. There are a large # of mouse strains and phenotypes to keep track of in this study. To improve readability and comprehension, I would suggest the authors summarize their findings in a supplementary table.

Reviewer #3: Hwang et al. presents a detailed genetic study examining the relative contributions of GliA and GliR in the Hh-dependent patterning of several tissues: the neural tube, limbs, and calvarium. The manuscript is very well-written and reader-friendly. While the topic of tissue patterning by differential regulation of GliA/GliR rations has certainly been examined previously, the genetic epistasis experiments presented here have generated new insights and led to a fairly simple model to explain how tissues respond differently to either different thresholds or different rations of GliA and GliR. This work generally seems very sound and merits publication in PLoS Genetics. I have only fairly minor points:

I appreciate that the authors quantified their results both for neural patterning and bone mineralization, but some methodological description of how these measurements were taken would be helpful. I don’t see this in the Methods section.

I was intrigued by the fact that while most of the tissues examined use the ratio sensing mode in their Hh response, some cell types use the threshold modes. I wonder if the authors think their model could be used to make predictions about the mode of response in other cell types, and how this might relate to cilia. For example, in cell types like the fibro-adipogenic progenitors described in Kopinke et al.(2017), where the presence of a cilium is suppressing Hh signaling.

**Have all data underlying the figures and results presented in the manuscript been provided?**

Reviewer #1: **No: **Several of the figures do not include information about n's for each genotype and in one instance, phenotype variability. The specific instances are noted in my review.

Reviewer #2: Yes

Reviewer #3: Yes

PLOS authors have the option to publish the peer review history of their article (what does this mean?). If published, this will include your full peer review and any attached files.

Reviewer #1: No

Reviewer #2: **Yes: **Benjamin R. Myers

Reviewer #3: No

---

## [Editor Report · Decision Letter 1]

24 Oct 2023

Dear Dr Mukhopadhyay,

We are pleased to inform you that your manuscript entitled "Context-dependent ciliary regulation of hedgehog pathway repression in tissue morphogenesis" has been editorially accepted for publication in PLOS Genetics. Congratulations!

Yours sincerely,

Gregory J. Pazour

Guest Editor

PLOS Genetics

Gregory Barsh

Editor-in-Chief

PLOS Genetics

Comments from the reviewers (if applicable):

**Data Deposition**

http://datadryad.org/submit?journalID=pgenetics&manu=PGENETICS-D-23-00769R1

**Press Queries**

---

## [Editor Report · Acceptance letter]

2 Nov 2023

PGENETICS-D-23-00769R1 

Context-dependent ciliary regulation of hedgehog pathway repression in tissue morphogenesis 

Dear Dr Mukhopadhyay, 

We are pleased to inform you that your manuscript entitled "Context-dependent ciliary regulation of hedgehog pathway repression in tissue morphogenesis" has been formally accepted for publication in PLOS Genetics! Your manuscript is now with our production department and you will be notified of the publication date in due course.

With kind regards,

Anita Estes

PLOS Genetics

On behalf of:
